Gut microbiota alterations in postmenopausal women with osteoporosis and osteopenia from Shanghai, China

Ji Jiaqing 1
Gu Zhengrong 2
Li Na 3
Dong Xin 3
Wang Xiong 2
Yao Qiang 2
Zhang Zhongxiao 4
Zhang Li 2 782010269@qq.com
Cao Liehu 2 traumahu@163.com
1 Department of Orthopedics, Tongji Hospital, School of Medicine, Tongji University , Shanghai , China
2 Department of Orthopedics, Luodian Hospital , Baoshan District, Shanghai , China
3 School of Medicine, Shanghai University , Shanghai , China
4 Hongqiao International Institute of Medicine, Tongren Hospital, Shanghai Jiao Tong University School of Medicine , Shanghai , China
Oliveira Sonia
Electronic publication date: 2024 May 31
Publication date: 2024
Volume: 12
Electronic Location ID: e17416
Received 2023 Dec 21; Accepted 2024 Apr 28
Copyright: © 2024 Ji et al.
Copyright year: 2024
Copyright holder: Ji et al.
License: This is an open access article distributed under the terms of the Creative Commons Attribution License, which permits unrestricted use, distribution, reproduction and adaptation in any medium and for any purpose provided that it is properly attributed. For attribution, the original author(s), title, publication source (PeerJ) and either DOI or URL of the article must be cited.
License URL: https://creativecommons.org/licenses/by/4.0/

Keywords: Gut microbiota, Postmenopausal osteoporosis, 16S rRNA sequencing

Funding: Scientific Research Project of Shanghai Municipal Health Commission 202040372 The Shanghai Baoshan District Medical Health Project (21-E-14) The work was funded by the Scientific Research Project of Shanghai Municipal Health Commission (Grant No. 202040372). The Shanghai Baoshan District Medical Health Project (21-E-14) provided funding for the APC. The funders had no role in study design, data collection and analysis, decision to publish, or preparation of the manuscript.

==============================
Background

The importance of the gut microbiota in maintaining bone homeostasis has been increasingly emphasized by recent research. This study aimed to identify whether and how the gut microbiome of postmenopausal women with osteoporosis and osteopenia may differ from that of healthy individuals.

Methods

Fecal samples were collected from 27 individuals with osteoporosis (OP), 44 individuals with osteopenia (ON), and 23 normal controls (NC). The composition of the gut microbial community was analyzed by 16S rRNA gene sequencing.

Results

No significant difference was found in the microbial composition between the three groups according to alpha and beta diversity. At the phylum level, Proteobacteria and Fusobacteriota were significantly higher and Synergistota was significantly lower in the ON group than in the NC group. At the genus level, Roseburia, Clostridia_UCG.014, Agathobacter, Dialister and Lactobacillus differed between the OP and NC groups as well as between the ON and NC groups (p < 0.05). Linear discriminant effect size (LEfSe) analysis results showed that one phylum community and eighteen genus communities were enriched in the NC, ON and OP groups, respectively. Spearman correlation analysis showed that the abundance of the Dialister genus was positively correlated with BMD and T score at the lumbar spine (p < 0.05). Functional predictions revealed that pathways relevant to amino acid biosynthesis, vitamin biosynthesis, and nucleotide metabolism were enriched in the NC group. On the other hand, pathways relevant to metabolites degradation and carbohydrate metabolism were mainly enriched in the ON and OP groups respectively.

Conclusions

Our findings provide new epidemiologic evidence regarding the relationship between the gut microbiota and postmenopausal bone loss, laying a foundation for further exploration of therapeutic targets for the prevention and treatment of postmenopausal osteoporosis (PMO).

Introduction

Postmenopausal osteoporosis (PMO) is a metabolic bone disease characterized by a decrease in bone mass caused by estrogen deficiency (Watts et al., 2010), which develops insidiously in the postmenopausal female population until a fragility fracture occurs. In China, the prevalence of osteoporosis is approximately 20.6% in the female population over 40 years of age (Wang et al., 2021a). Fragility fractures in the hip or spine are common after minor trauma or during daily activities, resulting in pain, malformation, dysfunction, and even death. Hip fracture, the most serious complication of PMO, has a 17% mortality rate during the first year (Forsen et al., 1999). PMO imposes considerable financial and medical burdens on patients and the healthcare system due to its serious consequences and high morbidity.

The human gut, which contains approximately 10 trillion bacteria (Qin et al., 2010) and a vast taxon diversity of more than 1,000 species (Falony et al., 2016), is the body’s largest repository of microorganisms. The gut microbiota assists the host in many ways and is involved in processes including the digestion of complex dietary macronutrients, production of nutrients and vitamins, protection against pathogens, and immune system maintenance, all of which are essential for maintaining human health. Sex, diet, lifestyle, and drug use all have an impact on the composition and structure of the gut microbiota (Xu et al., 2017). Microbes then affect the microenvironment of the gut by regulating the integrity of the intestinal barrier and the inflammatory response (Jones et al., 2013; Surana & Kasper, 2014).

Accumulating evidence has shown that the gut microbiota is associated with physiological bone metabolism and a series of inflammatory or metabolic bone diseases (Xu et al., 2017). The gut microbiota mainly exerts its effect on bone remodeling primarily through immune regulation, the secretion of metabolites, and nutrient acquisition (McCabe, Britton & Parameswaran, 2015). Previous animal experiments have revealed that modulation of the gut microbiota could promote bone health and growth (Sjogren et al., 2012; Yan et al., 2016), and the results may provide a theoretical basis for new therapeutic strategies for osteoporosis. High-throughput sequencing technology has substantially accelerated the study of the connection between the composition of the gut microbiota and human health, enabling the high-resolution profiling of microbial community compositions and functions (Franzosa et al., 2015). Several epidemiologic studies have demonstrated a significant correlation between specific gut microbiota features and reduced bone mineral density (BMD) in elderly adults. However, the pathogenesis of PMO is completely different from that of senile osteoporosis (He et al., 2020), and our understanding of gut microbiota regulation of bone metabolism in individuals with PMO is still limited due to the inconsistent results of earlier studies (He et al., 2020; Palacios-Gonzalez et al., 2020; Rettedal et al., 2021). In this study, we investigated alterations in the gut microbiota of postmenopausal women with osteoporosis and osteopenia from Shanghai, China, using 16S rRNA gene high-throughput sequencing. The fecal samples from 94 postmenopausal females were divided into osteoporosis (OP), osteopenia (ON), and normal control (NC) groups for more precise analysis. We anticipate that the findings will reduce the gap in knowledge revealed by previous research, assist in our understanding of the role played by the gut microbiota in the occurrence and development of PMO, and provide a foundation for the discovery of novel microbes that may serve as potential probiotic candidates for osteoporosis prevention and treatment.

Materials and Methods

Participant enrollment and data collection

Postmenopausal women in this study were enrolled from Luo Dian Hospital affiliated with Shanghai University (Shanghai, China). This study was approved by the Ethics Committee of Luo Dian Baoshan District, Shanghai Hospital (LDYY-KY-2020-24). The inclusion criterion for patients was a postmenopausal status defined as the absence of menstrual periods for 12 consecutive months and an age of more than 45 years. The exclusion criteria included the following: (1) use of estrogen, antibiotics, or anti-osteoporosis medications within six months of the fecal sample collection; (2) disability or inability to care for themselves; and (3) disease history of cancer, metabolic bone disease, digestive system disease, kidney disease, psychiatric disease, hyperthyroidism, or hypothyroidism. From 10 December 2020 to 20 May 2022, 94 postmenopausal women were enrolled and subjected to dual-energy X-ray absorptiometry (DXA) (Primus, Koera). All participants provided written informed consent before enrollment. BMD values at skeletal sites of the lumbar spine (LS) and left femoral neck (FN) were obtained using DXA assessment. BMD was recorded as the ratio of bone mineral content (g) to bone area (cm2), and the data are expressed as g/cm2. The T score is the difference in BMD between persons of the same sex who are 20 to 40 years old, given as a standard deviation of a reference BMD distribution. The T score threshold was used to define three groups. Participants included in the study were divided into the NC group (n = 23) with a T score of ≥−1, the ON group (n = 44) with a T score between −1 and −2.5, and the OP group (n = 27) with a T score of −2.5 or less. Baseline data (age, body weight, body mass index (BMI)), fracture history, dietary habits, and disease histories were collected. BMI was calculated as weight (kg) divided by the square of height (m).

Sample collection, DNA extraction, and PCR

Within an hour of collection, fecal samples were frozen at −80 °C until further laboratory detection. The microbial genome was extracted with a TIANamp Stool DNA Kit (Tiangen, Beijing, China) according to the manufacturer’s instructions. The integrity of genomic DNA was assessed through agarose gel electrophoresis. A NanoDrop 1000 Spectrophotometer (Thermo Fisher Scientific, Waltham, MA, USA) was used to assess the purity and concentration of DNA extracts. The bacterial 16S ribosomal RNA gene V3-V4 region was amplified with the specific primer 341F/805R. PCR amplification was achieved on a SimpliAmp™ Thermal Cycler (Thermo Fisher Scientific, Waltham, MA, USA) with a Phusion kit (Apexbio Technology Inc., Shanghai, China). DNA extraction, PCR, and 16S rRNA gene sequencing were conducted by Apexbio Technology Inc., (Shanghai, China).

16S rRNA gene sequencing and quality control

PCR products were sequenced using the Illumina NovaSeq 6000 with the 2 × 250 bp paired-end method. The raw reads were filtered by several steps to remove low-quality reads. First, reads containing low-quality bases (mass value ≤ 15) of more than 40 bp were removed. Second, reads with N-bases over 10 bp were removed. Third, reads that overlapped with the adapter above 10 bp were removed. Finally, the chimera sequence was removed, and clean reads were obtained.

Analysis of the microbial community

Sequences were analyzed using Quantitative Insights Into Microbial Ecology 2 (QIIME 2) (Bolyen et al., 2019). DADA2 method (Callahan et al., 2016) in QIIME 2 (versoin 2020.06) was applied to de-noise sequences, generating amplicon sequence variants (ASVs), and the species annotation and taxonomic analysis were performed based on the silva-138-99 16S rRNA database (Quast et al., 2013) by QIIME 2. A Venn diagram was generated to visualize common and unique ASVs between groups by JVENN web page. Rarefaction curves based on observed ASVs were drawn by QIIME 2 to assess whether the sequencing data is sufficient and to estimate species richness. Taxa abundance was measured and plotted by R software (ggplot2 package). Microbial alpha diversity was assessed with Chao1 and ACE for community richness and with Shannon and Simpson for community diversity. The significance of differences in the alpha diversity indexes (Chao1, ACE, Shannon, and Simpson) was measured by the Kruskal‒Wallis test and was visualized as a box diagram using R software (phyloseq package). Principal coordinate analysis (PCoA) was performed to evaluate microbial beta diversity at the ASV level based on Bray‒Curtis distance using R software (phyloseq/vegan package). Permutational multivariate analysis of variance (PERMANOVA) was performed to assess the between-group differences in microbial communities using the Adonis function in the R package vegan with 9999 permutations. Analysis of similarities (Anosim) was performed to test whether the difference between groups was significantly greater than the in-group difference and the results were visualized by R software (phyloseq/vegan package). The Kruskal‒Wallis test was used for between-group comparisons of relative abundances at the phylum and genus levels, and R software (DESeq2 package) was used for visualization. Linear discriminant effect size (LEfSe) analysis, which combines the Kruskal‒Wallis test with linear discriminant analysis (LDA), was applied to further identify the size of significant differences in the abundance of specific taxa between groups. Differences between groups were considered significant only when the LDA value was >2 and the p value was <0.05. Spearman correlation analysis was applied to measure the correlations between microbiota abundance and bone mass measurements such as BMD and T score. Ridge regression analysis was used to solve the collinear problem. The bioinformatics analysis was performed by Apexbio Technology Inc. (Shanghai, China).

Functional pathway predictions

Biological functions of osteoporosis-related gut microbiota were predicted with Phylogenetic Investigation of Communities by Reconstruction of Unobserved States (PICRUSt) based on the Kyoto Encyclopedia of Genes and Genomes (KEGG) database according to 16S rDNA sequencing data. LEfSe analysis was performed to identify significantly different functional pathways. The significant difference in KEGG ortholog abundances between groups was defined as a false discovery rate (FDR)-corrected p value <0.05 and an LDA value >2.

Statistical analysis

SPSS version 23.0 and R version 4.0.2 were used for statistical analysis. One-way analysis of variance (ANOVA) was performed to compare the differences in variables conforming to a normal distribution between groups, and the results were expressed as the means ± standard deviations. The Kruskal‒Wallis test was used for variables not conforming to a normal distribution. The chi-square test was used for categorical data. A two-sided p value <0.05 was considered to indicate statistical significance. Post hoc Bonferroni correction was performed to determine pairwise differences.

Results

General characteristics of the participants

As shown in Table 1, data from 94 participants (23 NC cases, 44 ON cases, and 27 OP cases) were included in the analysis. Differences in bone density measurements (T score and BMD of the lumbar spine (L1–L4) and neck of femur) were confirmed. No significant differences were found in variables including age, body weight, BMI, fracture history, vegetarian diet, and the presence of hypertension and diabetes among the three groups.

Table 1 Characteristics of the participants in the study.

Participants, (n = 94)	NC (n = 23)	ON (n = 44)	OP (n = 27)	p-value	
Age (years)	57.8 ± 5.5	59.3 ± 6.5	61.7 ± 6.7	0.087	
Weight (kg)	60.0 ± 5.5	60.1 ± 7.6	57.6 ± 9.0	0.357	
BMI (kg/m2)	24.4 ± 1.9	23.7 ± 2.7	23.1 ± 2.9	0.255	
Fracture history, n	9	25	20	0.050	
Vegetarian diet, n	1	3	3	0.687	
Hypertension, n	4	5	6	0.414	
Diabetes, n	2	4	4	0.689	
LS T score	0.7 ± 1.2	−1.4 ± 0.7	−2.7 ± 0.9	<0.001	
FN T score	−0.2 ± 0.7	−1.5 ± 0.4	−2.1 ± 0.9	<0.001	
LS BMD (g/cm2)	1.2 ± 0.1	1.0 ± 0.1	0.8 ± 0.1	<0.001	
FN BMD (g/cm2)	0.93 ± 0.1	0.8 ± 0.1	0.8 ± 0.1	<0.001	
Note:

One-way ANOVA or Chi-square test were used to determine significance. The values represent mean ± S.D. or number of samples per group. BMI, body mass index; LS, lumbar spine 1-4; FN, femoral neck; BMD, bone mineral density.

Comparisons of gut microbial diversity between the three groups

After error correction, 16S rRNA gene sequencing data from 94 stool samples were obtained, yielding a total of 13,624,956 high-quality reads, with a mean of 144,946 sequences per specimen (ranging from 79,886 to 265,679). A total of 14,233 ASVs were obtained from the total high-quality reads. The fecal bacteria comprised 334 genera, 133 families, 78 orders, 32 classes, and 20 phyla in total (Table 2). Figure 1 presents a Venn diagram for the OP, ON, and NC groups. The results of rarefaction analysis showed that the curves in all samples were near saturation, which indicated that our sequencing data were sufficient to capture the majority of microbiota taxa. No significant differences in alpha diversity indexes (ACE, Chao1, Shannon, and Simpson) were observed between any of the groups (Fig. 1). Regarding beta diversity, the PCoA plot did not show an obvious separation between groups (Fig. 1). The first two principal coordinates explain 12.25% and 5.66% of the total variance, respectively. PERMANOVA (R2 = 0.029, p = 0.074) did not show a significant difference in microbiota composition between groups. ANOSIM revealed that the between-group difference was greater than the within-group difference, but the difference was not significant (R = 0.029, p = 0.149) (Fig. 1).

Table 2 Bacterial compositions in each group at different levels.

	Phylum	Class	Order	Family	Genus	Species	ASV	
OP	16	23	60	105	252	199	5,356	
ON	16	24	64	113	278	201	7,125	
NC	14	19	56	97	243	162	4,245	
Total	20	32	78	133	334	276	14,233	
Note:

The 16S rRNA gene sequencing is not sufficient to identify the microbiota at the species level. OP: osteoporosis, ON osteopenia, NC: normal control, ASV: amplicon sequence variants.

Figure 1 Observed ASVs, alpha diversity indexes, and beta diversity indexes.

(A) Venn diagram showing the unique and shared ASVs in the OP, ON, and NC groups. (B–E) Between-group comparisons of the alpha diversity of gut microbiota (ACE, Chao1, Shannon, Simpson). The results are presented in a box diagram, and p values were calculated by using the Kruskal‒Wallis test. (F) Results of principal coordinate analysis (PCoA) of bacterial beta diversity based on the Bray‒Curtis distance. The first two principal coordinates explain 12.25% and 5.66% of the total variance, respectively. (G) The results of ANOSIM based on the Bray‒Curtis distance. R > 0 indicated that the difference between groups was greater than that within groups, with a p value < 0.05 indicating statistical significance.

Taxonomic composition of the gut microbial community and identification of significant taxa differences

Firmicutes, Bacteroidetes, Proteobacteria, and Actinobacteria were the four most prevalent phyla, accounting for more than 95% of the gut microbiota. At the phylum level, the relative abundances of Proteobacteria and Fusobacteriota were significantly higher in the ON group than in the NC group, and Synergistota was higher in the NC group (Fig. 2). Twenty major genera and the top 10 abundant genera with differential abundances are illustrated in Fig. 3. The results indicated that the relative abundance of Lactobacillus was significantly higher in the ON group than in the NC group. On the other hand, the proportions of Agathobacter and Roseburia were significantly lower in the ON group than in the NC group, and the proportions of Roseburia, Clostridia_UCG.014, and Dialister were higher in the NC group than in the OP group. According to the LEfSe analysis, at the phylum level, only Synergistota was enriched in the NC groups (Fig. 2). At the genus level, Lactobacillus was enriched in the OP group, Erysipelotrichaceae_UCG.003 was enriched in the ON group, and Roseburia and Dialister were enriched in the NC group (Fig. 3).

Figure 2 Bacterial community abundance at phylum level of each group.

(A) Relative abundances of the major taxa at the phylum level in the OP, ON, and NC groups. (B) Pairwise difference bar plot of the bacterial community abundance at the phylum level (p value < 0.05). (C) LEfSe analysis at the phylum level. Only the taxa with a p value < 0.05 and LDA value > 2 are shown in the figure.

Figure 3 Bacterial community abundance at genus level of each group.

(A) Relative abundances of the major taxa at the genus level in the OP, ON, and NC groups. (B and C) Pairwise difference bar plot of the bacterial community abundance at the genus level (p value < 0.05). (D) LEfSe analysis at the genus level; only the taxa with a p value < 0.05 and LDA value > 2 are shown in the figure.

Correlations between gut microbiota abundance and BMD measurements

No associations were found between gut microbiota abundance and BMD measurements at the phylum level (p > 0.05) by Spearman correlation analysis. The abundance of the Dialister genus was positively correlated with BMD and T score at the lumbar spine (p < 0.05) (Table 3). After adjusting for age and BMI in ridge regression analysis, no associations between gut microbiota abundance and BMD measurements were found at major genera level.

Table 3 Spearman correlation analyses of gut microbiota abundance and BMD measurements at genus level.

Taxonomic level	BMD_FN.	BMD_LS	T_FN	T_LS	
g_Agathobacter	0.198	0.131	0.199	0.126	
g_Megamonas	−0.094	0.028	−0.090	0.039	
g_Dialister	0.178	0.248*	0.162	0.256*	
g_Erysipelotrichaceae_UCG_003	−0.006	0.108	−0.018	0.106	
g_Clostridia_UCG_014	−0.078	0.062	−0.075	0.061	
g_Lactobacillus	−0.055	−0.159	−0.049	−0.160	
g_Roseburia	0.145	0.018	0.137	0.010	
Note:

Estimates were expressed as correlation coefficient and statistical significance is indicated by *p value < 0.05. LS, lumbar spine 1–4; FN, femoral neck; BMD, bone mineral density; g, genus.

Microbial functional pathway prediction

A total of 25 KEGG metabolic pathways were identified to show significantly different microbiota abundances between the three groups (all corrected p < 0.05) (Fig. 4). Pathways relevant to amino acid biosynthesis, vitamin biosynthesis, nucleotide metabolism and protein export were enriched in the NC group. Metabolites degradation, especially benzoate-related metabolites, was enriched in the ON group, and carbohydrate metabolism was primarily enriched in the OP group.

Figure 4 KEGG-based pairwise LEfSe analysis of differences in functional pathways (LDA > 2, FDR p value < 0.05).

Discussion

Increasing evidence suggests that the dynamic stability of the intestinal microbiota is of crucial importance for the maintenance of bone metabolism (Xu et al., 2017). In females, sufficient levels of estrogen aid in promoting beneficial bacteria, suppressing pathogenic species, and sustaining both the diversity and homeostasis of the gut microbiota. However, after menopause, estrogen deprivation can result in dysbiosis of the microbiota, thereby leading to increased osteoclastic bone resorption through immune system and endocrine system. Therefore, exploring the specific alteration of the gut microbiota composition in PMO patients is of significant value for screening potential probiotics with anti-osteoporotic effects, thus providing promising therapeutic targets for PMO.

In this study, differences in gut microbiota composition were investigated among the OP, ON, and NC groups through high-throughput 16S rRNA gene sequencing. Our findings showed that there was no significant difference in the alpha diversity of gut microbiota among any groups, which aligns with certain earlier studies (Das et al., 2019; Rettedal et al., 2021), but contradicts others (Wang et al., 2017; He et al., 2020; Wang et al., 2022). PERMANOVA of beta diversity revealed that the microbial composition of the OP and ON groups was not significantly different from that of the NC group. This result is inconsistent with previous studies that concluded that osteoporotic and osteopenia bacterial communities were much more similar to each other than to healthy communities (He et al., 2020; Rettedal et al., 2021). The contradictory outcomes of alpha diversity and beta diversity between different studies might be attributed to the vast heterogeneity of cohorts, including age, sex, dietary habits, menstrual status, and sample size, and caution should be taken in interpreting these results.

We further identified the taxonomic differences in the gut microbiota at the phylum and genus levels. At the phylum level, Synergistota was significantly higher in the NC group than ON group. In contrast, Proteobacteria and Fusobacteriota were significantly higher in the ON group than NC group. Our findings are consistent with previous studies that indicated Proteobacteria abundance was negatively associated with bone mass (Guss et al., 2017; He et al., 2020). Furthermore, the prevalence of Proteobacteria was reported to be related to a high risk of microbial dysbiosis, metabolic disease, and inflammation (Shin, Whon & Bae, 2015; Rizzatti et al., 2017). Fusobacteriota and Synergistota made up a very low proportion of the gut microbiota, and no literature indicate a relationship between them and bone metabolism.

At the genus level, we analyzed inter-group differences of top 20 abundant genera. Four genera were found at lower levels in the OP or ON groups compared with the NC group including Roseburia, Clostridia_UCG.014, Agathobacter, and Dialister. Notably, we observed that all of them belonged to the Firmicutes phylum which was reported to be correlated with the BMD values in several studies (Li et al., 2019; Greenbaum et al., 2022; Wang et al., 2022). Among all genera analyzed, Roseburia was mostly related to bone health because it was enriched in the NC group through LEfSe analysis and decreased in both the ON and OP groups. This result may suggest that the reduction of Roseburia plays a role in both the onset and progression stages of osteoporosis. The findings were supported by the previous research which indicated positive correlation of Roseburia abundance with BMD in elderly people (Li et al., 2019). We speculated that the anti-osteoporotic function of Roseburia may be associated with its ability to produce short-chain fatty acids (SCFAs)SCFAs including propionate and butyrate (Koh et al., 2016). SCFAs are the byproducts of the microbial fermentation of dietary fibers in the colon and predominantly comprise formate, acetate, propionate, and butyrate. Recently, emerging evidence has confirmed the function of SCFAs as key mediators in bone turnover which ameliorate calcium absorption by lowering the pH of the intestinal lumen and preventing the formation of calcium complexes (Weaver, 2015). According to an in vivo experiment in mice, propionate and butyrate could exert a protective effect on bone mass through the inhibition of osteoclast differentiation and bone resorption by downregulating essential osteoclast genes such as TRAF6 and NFATc1 (Lucas et al., 2018).

Dialister was another potential anti-osteoporotic probiotic genus discovered in this study which was found to be enriched in the NC group and to decrease in the OP group. Furthermore, it was the only prevalent genus whose relative abundance was positively correlated with BMD and T score at the lumbar spine through Spearman correlation analysis. The result might also be explained by the finding that Dialister spp. could produce propionate via the succinate pathway (Aurigemma et al., 2018). However, a recent study showed that patients with primary osteoporosis had a significantly higher abundance of Dialister (Xu et al., 2020). Additionally, it was reported that Dialister abundance positively correlated with an increase in interleukin (IL)-6 levels, which would result in bone loss (Martinez et al., 2013). The contradiction may suggest that different species of Dialister could have opposing effects on bone metabolism. However, 16S rRNA gene sequencing is not sufficient to identify the microbiota at the species level (Ravi et al., 2018). Therefore, the role of Dialister in bone remodeling is controversial, and further research is essential to determine the specific effect of its species.

The abundance reduction of Clostridia_UCG.014 and Agathobacter were also found in the OP and ON groups respectively compared with the NC group. However, they were not enriched in the NC group according to LEfSe analysis. Actually, there is no previous research indicating that these two genera might have anti-osteoporotic properties. Compared to Roseburia and Dialister, there seems to be less evidence linking Clostridia_UCG.014 and Agathobacter to osteoporosis resistance. The potential clues lie in the fact that Clostridia_UCG.014 and Agathobacter are members of the Clostridia class, which is well-known producer of SCFAs (Rettedal et al., 2021). In addition, the abundance of Clostridia in feces was found to be strongly associated with nonovarian systemic estrogens (Flores et al., 2012), which might help to prevent postmenopausal bone loss.

On the other hand, we found two genera which might be associated with PMO including Lactobacillus and Erysipelotrichaceae_UCG.003. Interestingly, Lactobacillus is one of the most commonly recognized probiotics. An in vivo experiment in ovariectomized mice revealed that Lactobacillus rhamnosus GG significantly improved intestinal barrier integrity and completely protected mice against sex steroid depletion–induced bone loss (Li et al., 2016). Lactobacillus reuteri was also reported to reduce basal TNF-α mRNA levels in the small intestine of male mice and increase bone density in the distal femur metaphyseal region and lumbar vertebrae (McCabe et al., 2013). Nevertheless, the findings on the relationship between Lactobacillus genera and osteoporosis in the human body remain inconsistent among different studies (Das et al., 2019; Li et al., 2019; Wei et al., 2021). Wei et al. (2021) suggested that different Lactobacillus species and strains may have distinct impacts on bone metabolism.

Erysipelotrichaceae_UCG.003 belongs to the Erysipelotrichaceae family, which was found to be linked to vitamin D insufficiency in postmenopausal individuals in a Mexican study (Palacios-Gonzalez et al., 2020). Since bone tissue mineralization is influenced by vitamin D levels, we speculate that Erysipelotrichaceae_UCG.003 in postmenopausal women might have the potential to contribute to an increased risk of bone loss although the relationship between Erysipelotrichaceae_UCG.003 and PMO is weak because it was only found to be enriched in the ON group.

Functional predictions based on KEGG pathways for gut microbiota were performed to explore potential metabolites that may be related to estrogen-reduced bone loss. In general, our results revealed that amino acid biosynthesis, vitamin biosynthesis, and nucleotide metabolism were enriched in the NC group, and all these elements might be essential in bone formation. In contrast, carbohydrate metabolism was primarily enriched in the ON and OP groups, and the findings are consistent with Rettedal’s et al. (2021) research. Although the results of the functional pathway prediction may provide a possible metabolic bridge between gut microbiota and bone loss, the interpretation should be treated with caution since no serum and fecal metabolite examinations were performed to confirm our assumption.

There are several advantages to our study. First, we focused on postmenopausal women in Shanghai, where individuals may have similar eating habits, to minimize the influence of diet and to exclude the effects of sex and menstrual status. Second, strict inclusion and exclusion criteria ensured that participants who had been exposed to factors that might affect bone density and the composition of the gut microbiota were excluded. Third, ASV analysis was first performed to reduce the impact of sequencing errors in this study, which provided more identification (99%) of gut microbiota than the operational taxonomic unit (OTU; 97%) analysis used in previous studies. Limitations should also be acknowledged. First, studies on the relationship between gut microbiota and osteoporosis has been conducted in various regions worldwide, and some studies have larger sample sizes than ours (Das et al., 2019; Greenbaum et al., 2022; Yang et al., 2022; Lin et al., 2023). However, the sample size of our study approached that of most similar studies (Rettedal et al., 2021; Li et al., 2019; He et al., 2020; Palacios-Gonzalez et al., 2020; Wei et al., 2021; Xu et al., 2020; Wang et al., 2022). In addition, our study represents the first investigation of postmenopausal women in the Yangtze River Delta region of China, one of the most populous regions in the country and characterized by a severely aging population. We believe that conducting research in different populations remains significant because it has the potential to validate previous research findings and uncover new targets not identified in earlier studies. Second, we did not detect metabolites like SCFAs or estrogens in feces and blood circulation. Therefore, we cannot determine the metabolites changes in PMO and analyze their relationship with gut microbiota composition changes. Third, due to the cross-sectional design of this study, a causal link between changes in intestinal bacteria and PMO cannot be determined.

Conclusions

In summary, we discovered taxa-specific variations in gut microbiota profiles of postmenopausal women with ON, OP, and NC. Based on the findings, we believe that Roseburia is highly likely to emerge as a promising anti-osteoporotic target. Dialister, Clostridia_UCG.014, and Agathobacter are also potential candidate microbiota with anti-osteoporotic properties, but their priority is lower than that of Roseburia. We hope that further animal interventional studies will help to confirm the causal relationship between the identified microbiota taxa and PMO and elucidate the underlying molecular mechanism.

Supplemental Information

Supplemental Information 1 STROBE checklist.

Supplemental Information 2 Raw data for participants.

Supplemental Information 3 Bioinformatics analysis.

We are grateful to all the participants for their contributions to this work. We would like to thank Apexbio Technology Inc. (Shanghai, China) for their technological assistance and AJE for English language editing.

Additional Information and Declarations

Competing Interests

Author Contributions

Human Ethics

Data Availability

The authors declare that they have no competing interests.

Jiaqing Ji conceived and designed the experiments, performed the experiments, analyzed the data, prepared figures and/or tables, authored or reviewed drafts of the article, and approved the final draft.

Zhengrong Gu conceived and designed the experiments, prepared figures and/or tables, authored or reviewed drafts of the article, and approved the final draft.

Na Li performed the experiments, authored or reviewed drafts of the article, and approved the final draft.

Xin Dong conceived and designed the experiments, analyzed the data, authored or reviewed drafts of the article, and approved the final draft.

Xiong Wang performed the experiments, authored or reviewed drafts of the article, and approved the final draft.

Qiang Yao performed the experiments, authored or reviewed drafts of the article, and approved the final draft.

Zhongxiao Zhang analyzed the data, authored or reviewed drafts of the article, and approved the final draft.

Li Zhang performed the experiments, authored or reviewed drafts of the article, and approved the final draft.

Liehu Cao conceived and designed the experiments, analyzed the data, authored or reviewed drafts of the article, and approved the final draft.

The following information was supplied relating to ethical approvals (i.e., approving body and any reference numbers):

This study was approved by the Ethics Committee of Luodian Hospital, Baoshan District, Shanghai, (LDYY-KY-2020-24) and conducted in compliance with relevant guidelines and regulations.

The following information was supplied regarding data availability:

The raw sequence data is available at GenBank: PRJNA1043868 and at figshare: Ji, Jiaqing (2023). 16s RNA sequencing data of gut microbiota in postmenopausal women from shanghai, china. figshare. Dataset. https://doi.org/10.6084/m9.figshare.24546388.v1.

https://www.ncbi.nlm.nih.gov/sra/PRJNA1043868.

The research software and data package are available in the Supplemental Files.

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
