# Peer review of "Gut microbiota alterations in postmenopausal women with osteoporosis and osteopenia from Shanghai, China"

_PeerJ, doi:10.7717/peerj.17416_

## Round 0.1 · original submission · Major Revisions

The authors should make some modifications before this manuscript can be sent for reviiew:

- prove the validity of data / statistical analysis with such low numbers of participants (27 individuals with osteoporosis (OP), 44 individuals with osteopenia (ON), and 23 normal controls (NC))

- this paper seems rather identical in terms of concept and general structure and even methods but it is not cited or in the bibliography: "Wang, H., Liu, J., Wu, Z. et al. Gut microbiota signatures and fecal metabolites in postmenopausal women with osteoporosis. Gut Pathog 15, 33 (2023). https://doi.org/10.1186/s13099-023-00553-0” - why? How is this different? What is the knowledge gap and how are they addressing it? - because there is significant literature relating microbiota and bone metabolism.

---

## Round 0.2 · Minor Revisions

Dear authors, thank you for your hard work. Please note that your manuscript still requires some revisions. I highlight in particular the importance of the detail in the material and methods along with all the necessary supplemental materials so that it can be reproduced by others if they wish and ensures transparency.
Please, refer to the reviewers' comments.

Reviewer 1 ·

Basic reporting

1.Line293-297 mentions that Proteus in NC group is significantly higher than that in ON group, which conflict with the following conclusion that Proteus is negatively correlated with bone mass.
2. Rigirously, the description of Line 369-371 is wrong, ASV reflects 99% similarity, not 100%.
3. In the picture part, there are some low-resolution image, and some words cannot be recognized. In addition, the author needs to confirm that whether the path in the first row of Figure 4 is fully displayed.

Experimental design

1.In the Introduction section, Line 93-94, the author believes that the sample size of the author's study is larger than that of similar studies. However, as shown in the literature listed in Q1, Lin X et al. included 517 perimenopausal/postmenopausal Chinese women (doi: 10.1038 / s41467-023-42005 - y), Yang and others into X 132 postmenopausal women (doi: 10.3389 / fimmu. 2022.930244), the author why not compare with the two articles? The authors do not appear to have read extensively the latest research on gut microbiota and postmenopausal osteoporosis.
2. For Participant Enrollment and Data Collection, why are there only detailed exclusion criteria but no detailed inclusion criteria?
3.For General Characteristics of the Participants, the including of chronic diseases, with no significant difference between the three groups ,should be explained in detail.

Validity of the findings

1.There have been many studies on intestinal flora and postmenopausal osteoporosis, so what is the innovation of this manuscript? (eg: ①Lin X, Xiao HM, Liu HM, et al. Gut microbiota impacts bone via Bacteroides vulgatus-valeric acid-related pathways. Nat Commun. 2023 Oct 27; 14(1):6853.②Lorenzo J. From the gut to bone: connecting the gut microbiota with Th17 T lymphocytes and postmenopausal osteoporosis. J Clin Invest. 2021 Mar 1; 131(5):e146619.③Yang X, Chang T, Yuan Q, Wei W, Wang P, Song X, Yuan H. Changes in the composition of gut and vaginal microbiota in patients with postmenopausal osteoporosis. Front Immunol. 2022 Aug 12; If 30244. )
2.In the Discussion part, the author seems to have written an overview. They should analyze and discuss their own results, provide valuable experiences or conclusions, and conduct a literature review, focusing on new and important findings from the research. The discussion should focus on the sub-focus, clear and hierarchical, and the text should strive to be concise and clear.
3.The association between intestinal flora and disease does not represent causality, and there is a lack of interventional studies to clarify the role of intestinal flora in disease.

Reviewer 2 ·

Basic reporting

The manuscript showns interestinf results on the gut microbiome in postmenopausal women with osteoporosis and osteopenia provides valuable insights into the relationship between gut microbiota and bone health. The thorough analysis of microbial composition and the identification of specific differences between groups contribute significantly to our understanding of postmenopausal bone loss.

Experimental design

I missed a detailed description of how the bioinformatics analyzes were carried out. I suggest adding it to the supplementary material
Have the sequences been deposited in a database?

Validity of the findings

I leave a question for the authors to reflect on. Discuss the sample size as a limitation of the manuscript.

Additional comments

No comment

Annotated reviews are not available for download in order to protect the identity of reviewers who chose to remain anonymous.

---

## Round 0.3 · accepted · Accept

Thank your for your re-submission. We all know how fast things can change when we are conducting research and someone can overtake our main project ideas. So i commend your perseverance.

Additionally, please include in the main manuscript the citation to the supplemental(s) namely the ApexBio Tech "protocol", as well as "raw data are available at NCBI GEO, accession numbers: PRJNA1043868 (Available for review at https://doi.org/10.6084/m9.figshare.24546388)". These should figure in the published version .

Additionally, you can do a better job at discussing the limitations or not of your study. (e.g. ". First, the sample size might be not large enough compared with latest research (Greenbaum J et al., 2022; Yang X et al., 2022; Lin X et al., 2023). ... what would be a large enough sample? how would it be more representative? How does your sample differ from others? How is it important for research on the topic? In a way, you can rephrase and include the arguments you used to answers the question "There have been many studies on intestinal flora and postmenopausal osteoporosis, so what is the innovation of this manuscript? ".

And on the inclusion criteria you can also do better (e.g. you simply say "Inclusion criteria: Postmenopausal women." - how was that diagnosis performed? cutoff values of the parameters/methods used to established diagnosis?... etc).

So, i am accepting this work for publication, with the condition of this minor improvements! Best of luck!